# Response Inhibition, Cognitive Flexibility and Working Memory in Obsessive-Compulsive Disorder, Generalized Anxiety Disorder and Social Anxiety Disorder

**DOI:** 10.3390/ijerph18073642

**Published:** 2021-03-31

**Authors:** Ana Isabel Rosa-Alcázar, Ángel Rosa-Alcázar, Inmaculada C. Martínez-Esparza, Eric A. Storch, Pablo J. Olivares-Olivares

**Affiliations:** 1Department of Personality, Assessment & Psychological Treatment, University of Murcia, Espinardo, 30100 Murcia, Spain; airosa@um.es (A.I.R.-A.); inmaculada.me@gmail.com (I.C.M.-E.); 2Department of Psychology, Catholic University of Murcia, Guadalupe, 30107 Murcia, Spain; aralcazar@ucam.edu; 3Department of Psychiatry and Behavioral Sciences, Baylor College of Medicine, Houston, TX 77030, USA; storch@bcm.edu

**Keywords:** obsessive-compulsive disorder, generalized anxiety disorder, social anxiety disorder, executive function

## Abstract

This study analyzed response inhibition, cognitive flexibility and working memory in three groups of patients diagnosed with obsessive-compulsive disorder, social anxiety disorder and generalized anxiety disorder, considering some variables that may influence results (nonverbal reasoning, comorbidity, use of pharmacotherapy). Neuropsychological measures were completed using a computerized Wisconsin card sorting test, Stroop color word test, go/no-go task, digits and Corsi. Significant differences were obtained among groups in cognitive flexibility and working memory variables. The obsessive-compulsive disorder (OCD) group showed the worst results. The social anxiety disorder group obtained greater effect sizes in visuospatial memory. However, significant differences between groups in visuospatial memory were no longer present when nonverbal reasoning was controlled. Comorbidity influenced interference in the OCD and generalized anxiety disorder (GAD) groups. In addition, the executive functions were differently influenced by the level of obsessions and anxiety, and the use of pharmacotherapy. Study limitations include a non-random selection of participants, modest sample size and design type (cross-sectional). The OCD group showed the worst results in flexibility cognitive and verbal working memory. Comorbidity, use of pharmacotherapy and level anxiety and obsessions were variables influencing the performance of executive functions.

## 1. Introduction

In the latest edition of the Statistical Manual of Mental Disorders [1], obsessive-compulsive disorder (OCD) is no longer considered an anxiety disorder. For this reason, it is assigned to a different chapter than other anxiety disorders such as social anxiety disorder (SAD) and generalized anxiety disorder (GAD). Despite these three diagnostic categories being considered independent categories, in practice there is high overlap in their respective diagnostic criteria [2]. It is therefore necessary to deepen the knowledge of these variables that allow for effective diagnosis. In this regard, some authors consider that the study of executive functions could be relevant [3].

EF (executive function) can be considered as a set of high-level mechanisms whose purpose is the regulation of cognitive, motor and psychophysiological responses to achieve an objective [4]. Response inhibition (RI), working memory (WM) and cognitive flexibility (CF) are considered the main mechanisms of EF [5,6].

RI integrates all neural circuits responsible for intentional and voluntary control that prevent interference of non-relevant information with ongoing responses or response patterns. This system can suppress information that was needed in the past but is no longer needed today [7]. RI is not considered a unitary EF; it can be divided into motor RI (or behavioral) and cognitive RI. Tests used to evaluate RI were go/no-go task for motor RI and the Stroop color and word test for cognitive RI. WM can be defined as all information retained or manipulated during a short period of time when it is no longer available in the environment. It implies the short-term storage of this information in order be used in other cognitive processes [8]. WM can be segregated into two components: verbal and visuospatial. Tests used for their evaluation were the digit test (Weschler) and the Corsi block task, respectively. CF is the ability that allows changing representation based on incoming information, and maintaining representation intact when changes are irrelevant [9].

Results obtained by some studies that analyzed EF in OCD, SAD and GAD are not conclusive. Along with OCD, the most researched anxiety disorders regarding EF are GAD, posttraumatic stress disorder (PTSD) and SAD [10]. OCD and PTSD are those most studied, with few studies on the other anxiety disorders.

In the successful review carried out by O’Toole and Pedersen [11], a total of 30 neuropsychological studies in patients diagnosed with SAD were compared. From all studies dealing with executive functions, only one found differences between groups diagnosed with SAD and healthy control (HC) groups [12]. From a total of five studies, two used the Wisconsin card sorting test (WCST) to measure set-shifting [13,14]; in none of these were differences found among patients with SAD and HC.

On the other hand, research developed by Fuji et al. [15] concluded that the level of anxiety in a sample of generalized social anxiety disorder (GSAD) correlated directly with the perseverative errors measured by the Wisconsin card sorting test (WCST). Along the same lines, regarding the existence of differences between HC and patients with SAD but from an anatomical–functional approach, the work carried out by Frick et al. [16] provides evidence of the negative correlation between the score on the Liebowitz Social Anxiety Scale [17] and cortical thickness in the right rostral anterior cingulate cortex (ACC).

The most recent studies about SAD also present inconsistent data and results. Heeren et al. [18] in a study of attention processing concluded that the functioning of executive networks is not altered. However, other authors find greater sensitivity to interference with memory tasks in SAD patients than in HC [19]. From a transdiagnostic point of view, Demetriou et al. [20] analyzed the functioning of executive functions among SAD, autism and early psychosis in young adults. Their results showed that deficits in these executive functions are not the primary contributors to disability for SAD. From an applied point of view, Zhao et al. [21] provided evidence that memory training is effective in treating GSAD in young adults.

Neuropsychological studies comparing executive functions among groups diagnosed with OCD, SAD and/or GAD are less frequent than those comparing a single type of disorder to HC. Cohen et al. [12] observed significant differences among patients with OCD and SAD in cognitive flexibility assessed with Trail Marking Test B—TMT B [22]. In the same vein, Kim et al. [23] reported that the OCD and GAD groups performed worse than the HC groups. Unlike these results, other studies found no difference between different anxiety disorders and performance in executive functions [24,25]. The results of the different studies available seem to indicate that groups with SAD show more visuoconstructional difficulties and to some degree less visual scanning ability than the rest of the disorders [11], which is striking, since in the literature this cognitive deficit is more associated with OCD [26,27]. When comparisons are made between SAD and GAD groups, no differences are found either in results of the emotional Stroop test, detection of neutral words or recalling neutral words [24,25,28].

Some authors highlight limitations affecting the interpretation of study results. Some studies use self-report measures rather than clinical interviews; others have not evaluated anxiety and depression with quantitatively validated measurement instruments; not all studies controlled for modulatory variables; some variables could influence results—OCD subtypes, use of pharmacotherapy, age and sex, duration of disorder, age of onset and comorbidity—all of which should be controlled in each study. The use of non-standardized neuropsychological tests prevents a comparison of results and replication. The application of dimensional measures versus a categorical approach may result in the generation of analogous groups, hindering a comparison of results [11,23,29,30,31,32]. A lack of consensus among the different neuropsychological studies in patients with OCD, SAD and GAD makes it necessary to further investigate the moderating variables related to neuropsychological function.

Aims were as follows: (1) To analyze RI, CF and WM differences among clinical groups. (1.1.) To study if EF performance can be influenced by nonverbal reasoning. (1.2) To analyze if there are differences in EF among clinical groups. (1.3) To assess the relationship between EF and obsessions, worry and social anxiety.

## 2. Materials and Methods

### 2.1. Participants

There were 89 participants between 17 and 61 years of age (Mean = 33.11, SD = 11.81), diagnosed with OCD, GAD and SAD [1]. Women comprised 55% of the sample. They were all Caucasian.

Inclusion criteria were as follows: (i) diagnosis of OCD, GAD and SAD [1], (ii) OCD participants were required to obtain ≥16 scores in Y-BOCS [33], (iii) GAD participants had to reach ≥56 in the Penn State Worry Questionnaire (PSWQ) [34], (iv) SAD participants had to obtain ≥52 scores in the Liebowitz Social Anxiety Scale (LSAS) [35]. Exclusion criteria for clinical groups were: (i) comorbidity with bipolar disorder, schizophrenic spectrum disorders and other psychotic disorders, personality disorders, anorexia, bulimia, disorders related to substance and addictive dependence and neurocognitive disorders. (ii) being under 15 or over 65 years of age. Recruitment is shown in Figure 1. Sample characteristics are presented in Table 1.

### 2.2. Procedure

The study met ethical standards of the Declaration of Helsinki and has been approved by the Ethics Committee of the University of Murcia (Spain). All families provided written informed consent.

After providing written consent, participants engaged in an individual diagnostic interview based on DSM-5 criteria, conducted by three clinical psychologists. An assessment was provided in two 60-min sessions by three experienced clinical psychologists who were trained by the fourth author for two sessions of one hour each. The test presentation order was the same for all participants. Participation in this study was voluntary and participants could discontinue at any given time.

### 2.3. Measures

#### 2.3.1. Clinical Measures

Yale Brown Obsessive-Compulsive Scale (Y-BOCS) [33]. Comprises 10 items assessing severity of OCD. It contains two subscales, obsessions (range = 0–20) and compulsions (range = 0–20) and a total score (range = 0–40). The scale has high internal consistency (*α* = 0.87–90) and good convergent validity (*r* = 0.74 to *r* = 0.47). A total average greater than or equal to 16 is considered of clinical significance. In this study, Cronbach’s alpha in the OCD group was 0.86.Penn State Worry Questionnaire (PSWQ) [34]. A 16-item self-report scale that assesses the general tendency to worry, especially present in generalized anxiety disorder. The cut-off point for detection of generalized anxiety disorder is 56. It is shown to have good psychometric properties, correlation with other measures of anxiety being satisfactory, for example, the SAI-R, with a correlation of 0.76. Cronbach’s alpha was high in the GAD group (*α* = 0.96).Liebowitz Social Anxiety Scale (LSAS) [35]. A 24-item scale that measures fear and avoidance of social situations over the past week. It consists of 11 items relating to social interaction and 13 items related to public performance. Each item is rated on two 4-point Likert-type scales by a clinician who may ask questions to clarify the appropriate rating for a specific participant. A total average greater than or equal to 51 is considered of clinical significance. Cronbach’s alpha in the SAD group was 0.91.Beck-II Depression Inventory (BDI) [36]. A 21-item self-report scale that measures depression severity. Classification of scores was as follows: minimal (0 to 13), mild (14 to 19), moderate (20 to 28) and severe (>29). The internal consistency coefficient ranged between 0.87 and 0.89. Cronbach’s alpha in all participants was 0.91.Beck Anxiety Inventory (BAI) [37]. A 21-item self-report scale that measures degree of anxiety. Classification of scores was as follows: minimal (0 to 7), mild (8 to 15), moderate (16 to 25) and severe (26+). The internal consistency coefficients varied between 0.85 and 0.93. Cronbach’s alpha in all participants was 0.92.

#### 2.3.2. Neuropsychological Measures

Wisconsin card sorting test (WCST) [38]. Assesses CF or attentional change using a set of cards. The most important measures are: number of categories completed, perseverative responses, total errors, perseverative errors and non-perseverative errors. The T-score is used taking age and educational level into account. The psychometric properties of the WCST have been widely researched, and it is a valid and reliable instrument, with oscillating reliability coefficients between 0.39 and 0.72.Stroop color word test (SCWT) [39]. Assesses the ability to inhibit the automatic tendency to respond verbally and, therefore, control response to conflicting stimuli (words, colors, words / colors and interference). This test is a good measure of cognitive inhibition. Test-retest reliability was 0.85, 0.81, 0.69.Go/no-go task [40,41] Evaluates motor RI. It involves two stimuli (arrows of different colors and positions), one requiring a response (go), and one requiring no response (no-go). It has presented good convergent validity (*r* = 0.87).Digits span test (WAIS-IV) [42]. This test evaluates verbal WM and consists of three sub-tasks (forward, backward and increasing order). The most important measure is the maximum number of elements (Span) that the individual can remember short term in backward or increasing order. Test-retest reliability oscillated between 0.70 and 0.85.Corsi block task (WMS-III) [43]. Evaluates visuospatial WM and consists of a forward and backward task. The most important data is the SPAN backward. Test-retest reliability oscillated between 0.70 and 0.82.Reynolds intellectual screening test (RIST) [44]. It has its origin in the RIAS scales (Reynolds Intellectual Assessment Scales) comprising two subtests: guess (verbal subtest) and categories (nonverbal subtest). In this study, only categories measuring nonverbal abstract reasoning were used. As with RIAS, it maintains high test-retest reliability 0.84.

Reliability indexes of neuropsychological measures were carried out on all persons who participated in the study.

### 2.4. Data Analysis

Firstly, Chi-square and one-factor ANOVA were used to examine potential group differences in clinical and demographic variables. Subsequently, ANOVA analysis (Brown–Forsythe) and post-hoc comparisons (Tukey or Games–Howell) of EF were then carried out. A covariance analysis was performed when there were significant differences among groups in some variables considered influential in their performance. Independent samples tests (Kruskal Wallis H test) were performed within each clinical group, taking into account the comorbidity, educational level and use of pharmacotherapy. The Pearson correlation was used to analyze the relationship between variables. Cohen’s d were calculated to estimate the magnitude of between-groups differences. All participants were included in the analyses. SPSS Statistic 22.00 (IBM, Chicago, IL, USA) was used for statistical analysis.

## 3. Results

### 3.1. Group Equivalence

Groups were not equivalent in years of disorder duration (*p* = 0.001), comorbidity *p* < 0.001), psychiatric treatment (*p* < 0.001) or type of pharmacotherapy (*p* < 0.001). Differences were seen in the categories variable (*p* = 0.003). See Table 1.

### 3.2. A Comparison of Groups in CF, RI and WM

Comparisons between the OCD and GAD groups reached statistical significance in the following variables: total errors (*p* = 0.002), perseverative responses (*p* = 0.007), perseverative errors (*p* = 0.003), non-perseverative errors of WSCT (*p* = 0.023), errors of omission of Go/No-Go (*p* = 0.036), with the GAD group obtaining the best scores. The highest effect size (ES) was in total errors and perseverative errors.

Comparisons between the OCD and SAD groups reached statistical significance in the following variables: perseverative errors (*p* = 0.036), non-perseverative errors (*p* = 0.021), digits span backward (*p* = 0.038), Corsi span forward (*p* = 0.001), Corsi span backward span (*p* < 0.01), total Corsi (*p* = 0.001), with the SAD group obtaining the best scores. The highest ES was in perseverative errors, non-perseverative errors and Corsi (span backward and total).

Comparisons between the GAD and SAD groups reached statistical significance only in the following variables: errors of omission (*p* = 0.000), digits span backward (*p* = 0.010), Corsi span forward (*p* = 0.001), Corsi span backward (*p* = 0.010) and total Corsi (*p* = 0.036). The SAD group obtained the best scores. The GAD group achieved better results in errors of omission, and the SAD group presented greater effect sizes in digit span backward and Corsi. Table 2 shows the ANOVA results for all variables.

### 3.3. CF, RI and WM Controlling Nonverbal Reasoning

Since the variable (nonverbal reasoning) obtained statistically significant differences among groups, an analysis of covariance was carried out (see Table 3). This variable only influenced the following results: Corsi total (*p* = 0.026), being the differences between SAD and GAD (*p* = 0.007).

On the other hand, significant differences between groups no longer exist when controlling nonverbal reasoning in Corsi SPAN backward (*p* > 0.05). In total Corsi, only mean differences between the GAD and SAD groups (*p* = 0.007) were maintained, with the latter group achieving best results. Comparison between OCD and SAD did not achieve significant differences (*p* = 0.071).

### 3.4. Intragroup Comparisons Based on Comorbidity and the Use of Pharmacotherapy

The OCD group presented significant differences regarding comorbidity in errors of commission (*p* = 0.025), Stroop interference (*p* = 0.023), digit span backward (*p* = 0.022), total digit (*p* = 0.008) and total Corsi (*p* = 0.015), with higher performance in patients with no comorbidity. Pharmacotherapy type influenced the Corsi backward score (*p* < 0.01). Patients who were taking antipsychotics presented the worst scores.

GAD participants suffering from other comorbid disorders presented significant differences in number of errors (*p* = 0.034), Stroop interference (*p* = 0.049) and digit span forward (*p* = 0.023), with higher performance in patients who only presented GAD. The use of pharmacotherapy influenced variables: number of categories completed (*p* = 0.017), total errors (*p* = 0.045), non-perseverative errors (*p* = 0.001), errors of omission (*p* = 0.005), errors of commission (*p* = 0.041), Corsi span backward (*p* > 0.05) and total Corsi (*p* > 0.05). Participants not taking medication performed better.

No differences were found in the SAD group regarding comorbidity. The use of pharmacotherapy influenced in the number of categories completed (*p* = 0.014), total errors (*p* < 0.001), perseverative response (*p* < 0.001), non-perseverative errors (*p* = 0.004), perseverative errors (*p* < 0.001), Corsi span backward (*p* = 0.022) and total Corsi (*p* = 0.001) with higher performance in patients not taking medication.

### 3.5. Correlation between EF and Obsessions, Worry and Social Anxiety

The OCD group only presented significant relationships between scores in OCI-R and errors of commission (*r* = 0.46, *p* = 0.027) the higher the score in obsessions the higher number of errors.

The GAD group achieved significant correlations between the scores of the Penn State Worry Questionnaire and Stroop colors (*r* = −0.54, *p* = 0.003), Interference Stroop (*r* = −0.39, *p* = 0.042), Total Digit (*r* = −0.44, *p* = 0.023) and Corsi Span backward (*r* = −0.41; *p* = 0.035) with Worry a variable that negatively influenced results.

The SAD group presented significant relationships between scores in Social performance and total errors (*r* = −0.55, *p* = 0.003), perseverative responses (*r* = −0.68, *p* < 0.001), perseverative errors (*r* = −0.67, *p* < 0.001), non-perseverative errors (*r* = −0.47, *p* = 0.019), Digit Span forward (*r* = −0.79, *p* < 0.001), Digit Span backward (*r* = −0.83, *p* < 0.011), Digit Span increasing (*r* = −0.70, *p* < 0.001), Total Digit (p =−0.90, r < 0.001), Corsi Span backward (*r* = −0.418, *p* = 0.030) and Total Corsi (*r* = −0.69, *p* < 0.001). Therefore, the higher the average in Social performance, the worse the results.

Social interaction reached significant relationships with total categories (*r* = −0.46, *p* = 0.014), total errors (*r* = −0.74, *p* < 0.001), perseverative responses (*r* = −0.79, *p* < 0.001), perseverative errors (*r* = −0.79, *p* < 0.001), non-perseverative errors (*r* = −0.72, *p* < 0.001), digit span forward (*r* = −0.70, *p* = > 0.001), digit span backward (*r* = −0.89, *p* < 0.011), digit span increasing (*r* = −0.50; *p* < 0.001), total digit (*r* = −0.68, *p* < 0.001), and total Corsi (*r* = −0.77, *p* < 0.001). Therefore, the higher the score in social interaction the worse the performance.

## 4. Discussion and Conclusions

Our main aim was to analyze CF, RI and WM differences among patients with OCD, GAD and SAD.

One of the most frequently used tests for assessing the CF is the Wisconsin card sorting test (WCST). The factors most closely related to OCD dysfunctions that WCST assesses are the number of categories completed and perseverative errors [45]. No differences were found among groups in the number of categories completed; however, some were found in perseverative responses, total errors, perseverative errors and non-perseverative errors. The mean scores on all variables measured by the WCST always followed the same trend: higher in SAD, lower in OCD and scoring in the middle in GAD. The ES were high–moderate in comparisons of OCD with SAD and GAD, and low between GAD and SAD. This deficit is consistent with the symptoms of patients with OCD, characterized by a maladaptive pattern of repetitive and inflexible thinking and behavior. Although numerous cognitive dysfunctions have been reported [29], impaired CF appears to be one of the key features in understanding the neural basis of OCD. These results are consistent with other studies [46].

The relationship of RI with anxiety disorders is unclear. On the one hand, some studies relate it only to disorders sharing transdiagnostic characteristics with OCD, such as Tourette’s disorder and trichotillomania [47], but not with other anxiety disorders [48,49]. These studies would justify the non-inclusion of OCD and other related disorders in the category of anxiety disorders. In contrast, some studies have found no relationship between RI and OCD [50,51,52]. Other studies postulate that RI is a common feature of all anxiety disorders so that a high score on the errors omission/errors commission rate correlates positively with anxiety levels [53].

Our study provides empirical evidence of statistically significant differences among the three groups of disorders in errors of omission. In particular, the ES between OCD and SAD was small but high between SAD-GAD and moderate–high between GAD-OCD. These data indicate that OCD and SAD present a processing speed slower than GAD. Contrary to our hypothesis, we found no differences among groups in cognitive RI (Stroop interference) and motor RI (errors of commission; these results are consistent with those reported by Abramovitch [29]. Although there were no differences among groups, the ES for errors commission were moderate in comparison to OCD and SAD versus GAD. Participants with GAD had better motor RI scores than OCD and SAD. On the other hand, the ES of OCD and GAD versus SAD in Stroop interference (cognitive RI) were also moderate. Participants with SAD scored worse on this factor than those with OCD and GAD. These results could be derived from those reported by Carver, Johnson and Timpano [54], who propose that availability of cognitive resources, emotional reactivity and psychopathology influence cognitive functions to a greater extent and diagnostic entity to a lesser extent. In the future, the development of techniques for the study of the central nervous system will allow us to learn to what degree endophenotypes are a cause or effect of the disorder, and in addition, should clarify whether functional dependence between different kinds of inhibition implies similar neural mechanism [55].

Perhaps the most compelling statement regarding the relationship of WM is that provided by Johnson and Gronlund [56]: “Anxiety was uncorrelated with WM capacity”. Despite this statement, there are studies providing evidence to the contrary [57,58,59] or favorable [60,61]. Some studies reported improved WM in anxiety [62,63]. In spite of the variety of results and conclusions, there seems to be consensus that in some circumstances, anxiety measures are related to poorer performance on WM measures [64]. In the present study, we analyzed verbal work memory (VWM) and visuospatial work memory (VSWM). In terms of VWM, only significant differences were obtained in span backward. No differences were obtained in digit span forward, increasing span and scalar store. These results indicated that the main functions of VWM are preserved in the same way in the three studied groups, coinciding with those reported by Stopa and Clark [65]. Regarding VWM, there is a clear trend in results; although no differences are found among groups, the SAD group presented better results than the OCD and GAD groups. The ES was moderate and high in the OCD-SAD and GAD-SAD comparisons and low for OCD-GAD except for the increasing span variable. At the level of verbal memory, GAD and OCD are more similar disorders than SAD. In VSWM, the data trend became more evident to the extent that statistically significant differences were reached in all variables measured with the Corsi cube test. These differences were maintained even when controlling for influence of non-verbal reasoning on the total Corsi mean. ES increased and became extremely high in OCD-SAD, high in GAD-SAD and low in OCD-GAD. This is not consistent with other studies indicating that visuoconstructional difficulties are a characteristic that SAD does not share with other anxiety disorders [11], but is consistent with that reported by others [66,67,68]. Martoni et al. [67] indicated that deficits in VSWM in the OCD group were directly proportional to workload increase.

Our third objective was to analyze whether there were differences in executive function within each clinical group due to comorbidity and the use of pharmacotherapy. The results reported that OCD patients with other comorbidities achieved worse results in cognitive and verbal RI, and VWM; the GAD with comorbidity group presented worse results in cognitive and verbal RI. These results are congruent with those reported by Abramovitch et al. [29] and Rosa-Alcazár et al. [69].

Nevertheless, the use of pharmacotherapy influenced visuospatial WM, with better performance in patients who did not take pharmacotherapy. This finding suggests that some psychotropic drugs may have cognitive side effects. Medication intake influenced CF in the SAD group and RI in the GAD group [32,51]. This could be due to several reasons, such as that patients without medication could have previously been without medication and this may have affected their cognitive functioning in a permanent way, or the comparison between medicated versus non-medicated has not taken into account the type of drug.

Our final aim was to assess the relationship between EF and obsessions and worry and social anxiety. The OCD group only obtained correlations between OCI-R scores and errors of commission: the greater the number of obsessions, the greater the number of errors of commission. These results are congruent with those indicated by Abramovitch et al. [30]. The GAD group reached correlations between PSWQ and interference Stroop and total digit and Corsi span backward. The higher the PSWQ score, the worse the performance in the indicated EF. These data could indicate the presence of a possible endophenotype for GAD with lower scores in cognitive RI, while for OCD, the worst results would be found in motor RI. In the SAD group, the level of social anxiety correlated with total errors, perseverative responses, perseverative errors, non-perseverative errors, digit span forward, digit span backward, digit increasing, total digit and total Corsi, so that higher scores in social anxiety performed worse in these EF.

Results indicate that high social anxiety levels correlate with worse VWM and CF scores. The clinical implications of these results could be used to improve existing treatments. The inclusion of treatment techniques aimed at alleviating deficits in EF could increase the efficacy, efficiency and effectiveness of the cognitive-behavioral treatments that are currently first choice for these disorders. For Rosa et al. [69], one of the possibilities for improvement would be to introduce techniques to increase CF in patients with OCD and GAD. Zhao et al. [21] recently demonstrated that WM training is an effective component for treating GAD. The applied potential could be huge. Future studies will have to show whether the percentage of variance explained using these new techniques justifies their use.

This study has some limitations such as non-random selection of participants, the use of a single evaluation tool for each variable evaluated, the small sample size, which prevents us from carrying out an analysis of the different subtypes of obsessions/compulsions and design type (cross-sectional). In future studies, the relationship between the different components of the variables analyzed should be analyzed with a larger sample size in order to see what kind of interactions arise between them. [70]. Cognitive responses can be quantified with different instruments and tasks, if possible, with a larger sample size. This would allow us to carry out studies that go deeper into the subtypes of disorders. Longitudinal research is needed to better delineate the changes in patients over time, and whether these are even maintained after therapeutic interventions.

The main conclusions have been that the OCD group showed worse results in cognitive flexibility and working memory than the GAD and SAD groups. The SAD group presented better results in visuospatial working memory than the OCD and GAD groups. Comorbidity affected performance in cognitive and verbal RI in the OCD and GAD groups. The use of medication influenced visuospatial WM in all clinical groups. Furthermore, CF in the. SAD group and RI in the GAD group were affected by this variable. Finally, executive functions were differently influenced by the level of obsessions and anxiety.

## Figures and Tables

**Figure 1 ijerph-18-03642-f001:**
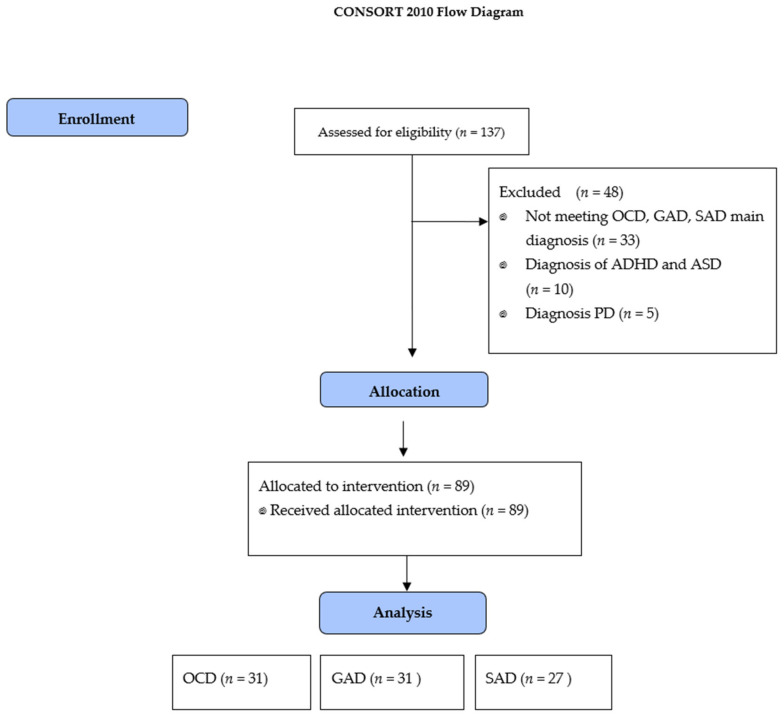
CONSORT flow diagrams of study development. OCD: Obsessive-compulsive disorder; GAD: Generalized anxiety disorder; SAD: Social anxiety disorder; ADHA: Attention deficit hyperactivity disorder; ASD: Autism spectrum disorder; PD: Personality disorders.

**Table 1 ijerph-18-03642-t001:** Sample measures.

Characteristics	OCD (*n* = 31)	GAD (*n* = 31)	SAD (*n* = 27)	*F/χ^2^*
Age (Mean ± SD)	35.51 ± 11.34	30.65 ± 11.18	33.29 ± 9.81	ns.
Sex *n* (%)				ns.
Men	15 (48.4)	11 (35.5)	14 (51.9)	
Women	16 (51.6)	20 (64.5)	13 (48.1)	
Years of disorder duration (Mean ± SD)	13.89 ± 11.71	4.85 ± 4.07	9.56 (8.62)	*F* (1.88) = 7.41; *p* = 0.001
Comorbidity *n* (%)				*χ*^2^ (2) = 19.22; *p* < 0.001
No comorbidity	15 (57.6)	13 (48.1)	27 (100)	
Comorbidity	16 (42.4)	18 (51.9)	0	
Marital status *n* (%)				ns.
Single	15 (48.4)	20 (64.5)	20 (74.1)	
Married	13 (41.9)	9 (29)	6 (22.2)	
Divorced	3 (9.7)	2 (6.5)	1 (3.7)	
Educational level *n* (%)				ns.
Elementary	7 (22.6)	5 (16.1)	3 (11.1)	
Secondary education	5 (16.1)	5 (16.1)	4 (14.8)	
High school	8 (25.8)	8 (25.8)	7 (25.9)	
University students	11 (35.5)	13 (42.0)	13 (48.2)	
Psychiatric treatment Yes	21 (67.7)	14 (45.2)	3 (11.1)	*χ*^2^ (2) = 24.13; *p* < 0.001
No	10 (32.3)	17 (54.8)	24 (88.9)	
Psychological treatment				ns
Yes No	24 (77.4) 7 (22.6)	26 (83.9) 5 (16.1)	27 (100) 0	
Type of pharmacotherapy				*χ*^2^ (6) = 32.22; *p* < 0.001
None Antidepressant	12 (38.7) 15 (48.4)	17 (54.8) 14 (45.2)	21 (77.8) 0	
Antipsychotic Antidepressant + antipsic.	1 (3.2) 3 (9.7)	0 0	6 (22.2) 0	
BAI (Mean ± SD)	19.55 ± 10.39	23.42 ± 13.92	25.81 ± 6.87	ns
BDI (Mean ± SD)	20.13 ± 11.42	24.25 ± 7.17	23.15 ± 8.35	ns.
Categories (Mean ± SD)	44.84 ± 12.26	49.61 ± 6.75	57.82 ± 6.79	*F* (2.88) = 6.87; *p* = 0.003

*n* = number; SD: Standard deviation; ns: Not significant; OCD: Obsessive-compulsive disorder; GAD: Generalized anxiety disorder; SAD: Social anxiety disorder.

**Table 2 ijerph-18-03642-t002:** ANOVA and effect size (ES) of the executive functions.

Measures	Dependent Variables	Group	*N*	MEAN	*SD*	*F*	*d* Cohen *	
WCST	Number of categories completed	OCD	31	4.90	1.47	*F* (2; 79.3) = 0.31, *p* > 0.05	OCD-GAD	−0.11
		GAD	31	5.09	1.90		GAD-SAD	−0.08
		SAD	27	5.22	1.25		OCD-SAD	−0.23
	Perseverative responses	OCD	31	40.16	11.09	*F* (2; 48.4) = 3,90, *p* = 0.027	OCD-GAD	−0.80
		GAD	31	47.93	8.03		GAD-SAD	−0.20
		SAD	27	50.03	12.42		OCD-SAD	−0.83
	Total errors	OCD	31	39.03	9.11	*F* (2; 49.7) = 3.90, *p* = 0.026	OCD-GAD	−0.89
		GAD	31	46.54	7.65		GAD-SAD	−0.25
		SAD	27	46.81	13.81		OCD-SAD	−0.68
	Perseverative errors	OCD	31	39.54	10.53	*F* (2; 40.1) = 5.00, *p* = 0.011	OCD-GAD	−0.89
		GAD	31	47.87	8.13		GAD-SAD	−0.23
		SAD	27	50.44	13.54		OCD-SAD	−0.91
	Non-perseverative errors	OCD	31	39.87	8.04	*F* (2; 52.8) = 5.04, *p* = 0.009	OCD-GAD	−0.69
		GAD	31	45.29	7.64		GAD-SAD	−0.34
		SAD	27	48.15	9.30		OCD-SAD	−0.96
Go/No-Go	Errors omission	OCD	31	1.29	2.39	*F* (2; 35.8) = 5,05, *p* = 0.012	OCD-GAD	−0.69
		GAD	31	0.16	0.37		GAD-SAD	1.51
		SAD	27	0.92	0.62		OCD-SAD	−0.20
	Errors commission	OCD	31	2.55	1.87	*F* (2; 75.0) = 2.30, *p* > 0.05	OCD-GAD	−0.44
		GAD	31	1.80	1.49		GAD-SAD	0.53
		SAD	27	2.48	1.01		OCD-SAD	−0.05
Stroop	Stroop interference	OCD	31	49.06	7.36	*F* (2; 69.3) = 1.55, *p* > 0.05	OCD-GAD	−0.05
		GAD	31	49.42	7.55		GAD-SAD	−0.49
		SAD	27	45.70	7.78		OCD-SAD	0.44
Digits	Span forward	OCD	31	6.42	1.28	*F* (2; 81.3) = 1.74, *p* > 0.05	OCD-GAD	−0.33
		GAD	31	6.83	1.18		GAD-SAD	−0.17
		SAD	27	7.03	1.14		OCD-SAD	−0.50
	Span backward	OCD	31	4.87	1.23	*F* (2; 81.8) = 5.02, *p* = 0.009	OCD-GAD	−0.14
		GAD	31	4.71	1.10		GAD-SAD	−0.79
		SAD	27	5.66	1.30		OCD-SAD	−0.63
	Span increasing	OCD GAD SAD	31 31 27	5.48 5.96 5.81	1.20 0.75 1.30	*F* (2; 70.8) = 1.51; *p* > 0.05	OCD-GAD GAD-SAD OCD-SAD	−0.48 0.16 −0.28
	Total digit	OCD	31	10.00	3.32	*F* (2; 73.1) = 2.85, *p* > 0.05	OCD-GAD	−0.09
		GAD	31	10.29	3.35		GAD-SAD	−0.49
		SAD	27	12.22	4.52		OCD-SAD	−0.57
Corsi	Span forward	OCD	31	10.67	3.11	*F* (2; 76.6) = 8.78, *p* < 0.01	OCD-GAD	−0.01
		GAD	31	10.70	2.03		GAD-SAD	−1.13
		SAD	27	13.11	2.24		OCD-SAD	−0.89
	Span backward	OCD	31	7.51	3.06	*F* (2; 63.6) = 10.46, *p* < 0.01	OCD-GAD	−0.28
		GAD	31	8.67	5.04		GAD-SAD	−0.77
		SAD	27	11.77	2.29		OCD-SAD	−1.56
	Total	OCD	31	16.13	3.13	*F* (2; 67.3) = 6.39, *p* = 0.003	OCD-GAD	−0.14
		GAD	31	16.77	5.88		GAD-SAD	−0.65
		SAD	27	20.00	3.60		OCD-SAD	−1.15

OCD: Obsessive-compulsive disorder; GAD: Generalized anxiety disorder; SAD: Social anxiety disorder. ES: Effect size. * Negative *Ds* indicated that the group compared in first place reached the worst score achieved by the group appearing in second place.

**Table 3 ijerph-18-03642-t003:** Executive functions performance with nonverbal reasoning as covariate.

Measure	Dependent Variable	Group	*N*	Adjusted Mean	*F*
Corsi	Total Corsi	OCD	31	16.96	*F* (2, 85) = 3.96, *p* = 0.026
		GAD	31	15.78	
		SAD	27	21.22	

OCD: Obsessive-compulsive disorder; GAD: Generalized anxiety disorder; SAD: Social anxiety disorder.

## Data Availability

The data presented in this study are available on request from the corresponding author. The data are not publicly available due to the research team continues to conduct experiments to increase the total data. When all the experimental work is completed, the complete database will be published.

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
