# Peer review of "Response Inhibition, Cognitive Flexibility and Working Memory in Obsessive-Compulsive Disorder, Generalized Anxiety Disorder and Social Anxiety Disorder"

_ijerph, 2021, doi:10.3390/ijerph18073642_

Round 1

Reviewer 1 Report

The present paper investigates the impact of different psychopathological conditions on measures of cognitive flexibility and working memory. In sum, I have to state that I found the paper quite hard to read, missed much relevant information (i.e., inferential statistics for pairwise comparisons), alignement of reliability measures to groups and dedicated, specific hypotheses for which effects to look. Below, in a manuscript based order I highlight my points with some examples.

 Comments based on manuscript:

page 2, line 60, please change to Wechsler

page 3, line 105, please exchange confounding by either moderating or mediating variable (which sort of incluence do you expect? a change in relationship or the emergence of  a relationsship )

page 3, line 107: related to comment on line 105, all three aims following aim one are sub-aims, I would very much appreciate specific , directly formulated hypotheses (i.e., higher WM in the visual spatial domain goes along with less perversance errors in the WCST) rather than the broad we investigated influences of x on y. There is enough literature out there to formulate specific hypotheses that allow for precise and informed testing and might therefore increase the theoretical value of the paper greatly.

Table 1 : did you run follow-up pairwise comparison to identify the deviant conditions? in case you did please report, in case you did not, please state why

clinical measures and assessed  neuropsychological tasks: sometimes you report reliability (likely for the whole sample?), sometimes you report several measures (group-based?), please be more precise what the different numbers reflect and whether they refer to all participants or a subgroup.

Finally it is and has never been sufficient to report p-values only without accompagnying test and inferential statistical values, especially when those reflect post-hoc tests to uncover main effects of groups detected in the one-way ANOVA.

Please make sure to introduce abbrevations, I assume ES reflects Effect Size but I am lost with MT (memory task?, page 5 line 183) - thank you.

Overall, as regards the result section it reflects the rather unprecise aim of the study and does really create confidence in the results as simply a bunch of tests are run on a rich amount of data (which should be made available on OSF or the like for interested colleagues). Thus, also I appreciate the amount of work the authors undertook, I just felt that the study had a higher potential and could with some more theorizing and adding some more literature on EF and psychopatholody (i.e., Meiran et al., 2007 or work by Todd Braver or Marie Banich and so forth on cognitive flexibility and inhibition, Frederik Verbruggen,  on stop signal or Diamond as a review whereby I spotted several developmental studies (Chevalier and also Salthouse) which might actually not be much suited for the present endeavour).

Finally, I would appreciate when citations rules are applied appropriately as i had a really hard time linking references to citations.

I apologize for my rather harsh commenting but hope to provide some suggesting of how to revise the paper to make it an important contribution.

Author Response

Dear reviewer,

First of all, thank you very much for the suggested contributions to this study as this allows us to make improvements. We appreciate your invaluable time.

 We have modified the areas suggested.

Faithfully,

Ana Isabel Rosa-Alcázar & Pablo José Olivares-Olivares

The present paper investigates the impact of different psychopathological conditions on measures of cognitive flexibility and working memory. In sum, I have to state that I found the paper quite hard to read, missed much relevant information (i.e., inferential statistics for pairwise comparisons), alignement of reliability measures to groups and dedicated, specific hypotheses for which effects to look. Below, in a manuscript based order I highlight my points with some examples.

 Comments based on manuscript:

page 2, line 60, please change to Wechsler

Authors: Thanks to the reviewer. The manuscript is corrected.

page 3, line 105, please exchange confounding by either moderating or mediating variable (which sort of incluence do you expect? a change in relationship or the emergence of  a relationsship )

Authors: Changed by moderating variable

page 3, line 107: related to comment on line 105, all three aims following aim one are sub-aims, I would very much appreciate specific , directly formulated hypotheses (i.e., higher WM in the visual spatial domain goes along with less perversance errors in the WCST) rather than the broad we investigated influences of x on y. There is enough literature out there to formulate specific hypotheses that allow for precise and informed testing and might therefore increase the theoretical value of the paper greatly.

Authors: The authors have made the following changes to objectives and sub-objectives 2), 3) y 4). The authors decided not to include hypotheses since the majority of scientific publications indicate objectives to be contrasted, the direction of the hypotheses being not so relevant. However, the authors considered that the OCD group would obtain worse results in all the variables measured, that comorbidity, use of medication and severity of symptoms would affect the results in a negative way.

Table 1 : did you run follow-up pairwise comparison to identify the deviant conditions? in case you did please report, in case you did not, please state why

Authors: The research team is currently working on follow-ups. The results will be published once the last follow-up (24 months) is completed.

clinical measures and assessed  neuropsychological tasks: sometimes you report reliability (likely for the whole sample?), sometimes you report several measures (group-based?), please be more precise what the different numbers reflect and whether they refer to all participants or a subgroup.

Authors: The reliability indices of the scales specific to each disorder were carried out with each clinical group, while those referring to common variables (BDI, BAI, neuropsychological measures) were calculated with all study participants. Modification is made to the manuscript.

Finally it is and has never been sufficient to report p-values only without accompagnying test and inferential statistical values, especially when those reflect post-hoc tests to uncover main effects of groups detected in the one-way ANOVA.

Authors: The authors consider that in order not to increase the number of tables the post-hoc analyses could be collected with the p-value and Cohen's d-value. The post-hoc comparisons were performed with Tukey or Games-Howel.

Please make sure to introduce abbrevations, I assume ES reflects Effect Size but I am lost with MT (memory task?, page 5 line 183) - thank you.

Authors: We have indicated on line 215 the acronym ES (effect size). On line 185 we have corrected the typo MT to WM (working memory).

Overall, as regards the result section it reflects the rather unprecise aim of the study and does really create confidence in the results as simply a bunch of tests are run on a rich amount of data (which should be made available on OSF or the like for interested colleagues). Thus, also I appreciate the amount of work the authors undertook, I just felt that the study had a higher potential and could with some more theorizing and adding some more literature on EF and psychopatholody (i.e., Meiran et al., 2007 or work by Todd Braver or Marie Banich and so forth on cognitive flexibility and inhibition, Frederik Verbruggen,  on stop signal or Diamond as a review whereby I spotted several developmental studies (Chevalier and also Salthouse) which might actually not be much suited for the present endeavour).

Authors: Thank you for your comments. The authors have reviewed and made minor revisions to the document:

“should clarify whether functional dependence between different kinds of inhibition implies similar neural mechanisms”[55]

This could be due to several reasons, such as that patients without medication could have previously been without medication and this may have affected their cognitive functioning in a way permanent, or the comparison between medicated vs non-medicated has not taken into account the type of drug”

“The relationship between the different components of the variables analysed should be analysed with a larger sample size in order to see what kind of interactions arise between them”[70]

Finally, I would appreciate when citations rules are applied appropriately as i had a really hard time linking references to citations.

Authors: We have reviewed the citations according to the format indicated by the journal. The correspondence between citations and references has been checked and is appropriate.

I apologize for my rather harsh commenting but hope to provide some suggesting of how to revise the paper to make it an important contribution.

Reviewer 2 Report

This is an interesting manuscrip on response inhibition, cognitive flexibility and working memory in three groups of patient diagnosed with Obsessive-Compulsive Disorder, Social Anxiety Disorder and Generalized Anxiety Disorder.

Even if I see promise here, I suggest to address the following points:

-Which are your hypothesis?

-Authors need to justify the age range. Why being under 15 or over 65 years of age is an exclusion criteria?

-Please justify the sample size, is it enough power?

-Did participants under 18 years old assent to participate in the study?

-Analysis should be readdressed. What assumptions have been satisfied to carry out ANOVA? Is age examined as a possible counfunder? I am not really convinced by pairing participants is enought. On the other hand, include your MSE and effect size on it. The last section seems not enough for the proposes of the study. Maybe a regression analysis could be of interest, where age is also included.

-What is the home message in your last section?

Author Response

Dear reviewer,

First of all, thank you very much for the suggested contributions to this study as this allows us to make improvements. We appreciate your invaluable time.

 We have modified the areas suggested.

Faithfully,

Ana Isabel Rosa-Alcázar & Pablo José Olivares-Olivares

This is an interesting manuscrip on response inhibition, cognitive flexibility and working memory in three groups of patient diagnosed with Obsessive-Compulsive Disorder, Social Anxiety Disorder and Generalized Anxiety Disorder.

 Authors: Thank you very much for your kind comments

Even if I see promise here, I suggest to address the following points:

-Which are your hypothesis?

Authors: The authors decided not to include hypotheses since the majority of scientific publications indicate objectives to be contrasted, the direction of the hypotheses being not so relevant. However, the authors considered that the OCD group would obtain worse results in all the variables measured, that comorbidity, use of medication and severity of symptoms would affect the results in a negative way

-Authors need to justify the age range. Why being under 15 or over 65 years of age is an exclusion criteria?

Authors: The upper limit of the range was set at 65 years to avoid including in the study people suffering from cognitive impairment affecting EF. The lower limit was 15 years as we considered that a lower age could affect the results as the disorder might not be sufficiently generalised and consolidated to generate evident changes in executive functions. In order to control for these mediating variables that could influence the results, it was decided to apply the indicated age range.

-Please justify the sample size, is it enough power?

Authors: We performed an a priori determination to anticipate the minimum sample size needed to be able to perform this analysis. For a power of 95%, an alpha level of 5%, for an inter factor of 3 groups and 3 dependent variables, and assuming a mean effect size (Cohen, 1988), f2 = 0.0625, less than 75 participants would be required. To safeguard these analytical conditions, sample sizes of around 25 participants per group (if possible, more) but always above 18 will be attempted. These calculations were performed with the statistical programme G*Power 3.1.9.2 (Buchner, Erdfelder, Faul, & Lang, 2014).

-Did participants under 18 years old assent to participate in the study?

Authors: All participants agreed to take part in the research. For minors, an informed consent form was prepared for their guardians to authorise their participation. All documentation was sent to the Ethics Committee of the University of Murcia.

-Analysis should be readdressed. What assumptions have been satisfied to carry out ANOVA? Is age examined as a possible counfunder? I am not really convinced by pairing participants is enought. On the other hand, include your MSE and effect size on it. The last section seems not enough for the proposes of the study. Maybe a regression analysis could be of interest, where age is also included.

Authors: Authors: All ANOVA assumptions were met. Normality, independence and homoscedasticity. We consider that we should not include the mean squared error (MSE) or mean squared deviation (MSD) as this is the format used in most published studies. We believe that we should not present a regression analysis due to the small sample size in each group. However, it is an element to be taken into account when we have increased the sample size. We therefore welcome your indications.

-What is the home message in your last section?

Authors: We did not quite understand the reviewer's indication. Our last section begins with a description of our general objective to try to put the reader in perspective before proceeding with the development of the section.

Reviewer 3 Report

Authors studied response inhibition, cognitive flexibility and working memory for patients with Obsessive-Compulsive Disorder, Social Anxiety Disorder and Generalized Anxiety Disorder. In this study, authors considered variables that may influence results such as nonverbal reasoning, comorbidity, use of pharmacotherapy. Neuropsychological Measures completed with computerized Wisconsin Card Sorting Test, Stroop Color Word Test, Go/NoGo Task, Digits and Corsi. Authors found significant differences among groups with cognitive flexibility and working memory variables. Social Anxiety Disorder group has greater effect sizes in visuospatial memory. Executive functions were differently influenced by level obsessions and anxiety and use of pharmacotherapy. Study limitations include a non-random selection of participants, modest sample size, and design type (cross-sectional).

Author Response

Dear reviewer,

First of all, thank you very much for the suggested contributions to this study as this allows us to make improvements. We appreciate your invaluable time.

 We have modified the areas suggested.

Faithfully,

Ana Isabel Rosa-Alcázar & Pablo José Olivares-Olivares

Authors studied response inhibition, cognitive flexibility and working memory for patients with Obsessive-Compulsive Disorder, Social Anxiety Disorder and Generalized Anxiety Disorder. In this study, authors considered variables that may influence results such as nonverbal reasoning, comorbidity, use of pharmacotherapy. Neuropsychological Measures completed with computerized Wisconsin Card Sorting Test, Stroop Color Word Test, Go/NoGo Task, Digits and Corsi. Authors found significant differences among groups with cognitive flexibility and working memory variables. Social Anxiety Disorder group has greater effect sizes in visuospatial memory. Executive functions were differently influenced by level obsessions and anxiety and use of pharmacotherapy. Study limitations include a non-random selection of participants, modest sample size, and design type (cross-sectional).

Authors: Thank you very much for your comments. We understand that the article has been of interest to you.

Round 2

Reviewer 1 Report

Thank you very much for revising the manuscript and taking into account some of my comments. Here are some final ones:

Whereas you now better describe the reliability analysis for the questionnaire this improved explanation is skipped for the neuropsychological tests employed for which still several values or ranges are reported but neither numbers are given which would be a more precise description than "adequate values were obtained".

However, I might have missed it in the first version, but I guess Cohen's d reported in Table 2 doe actually reflect the differences between groups and then the reported p values make sense as well. My apologies.

Please note that I did refer in my intial review to "follow - up tests", simply to tidentify deviant group but did not intend to include values from a follow-up study.

Still I am not quite pleased with the selected references but think this might be a problem of availability of the suggested papers and a stronger focus of clinical and not cognitive psychology.

Finally, I wondered on the addition of two more questionnaires that were not present in the first version and strongly encourage to make the data set publicly available for interested colleagues.

Author Response

Dear reviewer, we will reply in the same order:

- We have added the reliability indices for the neuropsychological measures that did not appear in the manuscript (row 190 and 193).

- Thank you for your comment, it is indeed as you describe. The footer of table 2 has been modified: “EN: effectsize. *Negative Ds indicated that the group compared in first place reached the worst score achieved by the group appearing in second place.” (row 127)

- If the reviewer considers that we should indicate something else in the manuscript, we would be grateful if you could specify what we should include, as we do not understand what you are trying to indicate. Thank you.

- Thank you for your consideration. Indeed the cognitive approach items you indicated we will incorporate them in future related studies.

- Dear reviewer, all questionnaires have been presented in all versions of the manuscript. We would be grateful if you could indicate which questionnaires you think are missing. We are currently continuing to expand the database and plan to make it public in the near future.

Reviewer 2 Report

Thank you for addressing most of my concerns. My only doubt is related to the age groups. As this variable is not controlled into group age, could authors explain if theya re interest in age as a cntinuos variable in the study?

Author Response

The authors are interested in controlling the age variable but at the present time the sample does not include a sufficiently large group of young patients to allow valid results to be obtained in the comparisons. In future studies the sample will be increased and the influence of the age variable will be controlled.

Thank you for your work and considerations.